# COVID-19 outbreaks in nursing homes: A strong link with the coronavirus spread in the surrounding population, France, March to July 2020

Muriel Rabilloud[1,2]*, Benjamin Riche[1,2], Jean François Etard[3,4], Mad-Hélénie Elsensohn[1,2], Nicolas Voirin[5], Thomas Bénet[6], Jean Iwaz[1,2], René Ecochard[1,2], Philippe Vanhems[7,8]

1 Équipe Biostatistique Santé, Laboratoire de Biométrie et Biologie Évolutive, CNRS UMR 5558, Université Lyon 1, Université de Lyon, Villeurbanne, France, 2 Service de Biostatistique-Bioinformatique, Pôle Santé Publique, Hospices Civils de Lyon, Lyon, France, 3 TransVIHMI, UMI 233 Institut pour la Recherche et le Développement, U 1175, Institut National de la Santé et de la Recherche Médicale, Université de Montpellier, Montpellier, France, 4 EpiGreen, Paris, France, 5 Epidemiology and Modelling in Infectious Diseases (EPIMOD), Dompierre-sur-Veyle, France, 6 Auvergne-Rhône-Alpes Regional Office, Santé Publique France, Lyon, France, 7 Service d'Hygiène Hospitalière, Épidémiologie, Infectiovigilance et Prévention, Hospices Civils de Lyon, Lyon, France, 8 Laboratoire des Pathogènes Émergents, Centre International de Recherche en Infectiologie (CIRI), Université de Lyon, Lyon, France

* muriel.rabilloud@chu-lyon.fr

**Data Availability Statement:** All relevant data are within the paper and its Supporting information files.

## Abstract

### Background

Worldwide, COVID-19 outbreaks in nursing homes have often been sudden and massive. The study investigated the role SARS-CoV-2 virus spread in nearby population plays in introducing the disease in nursing homes.

### Material and methods

This was carried out through modelling the occurrences of first cases in each of 943 nursing homes of Auvergne-Rhône-Alpes French Region over the first epidemic wave (March-July, 2020). The cumulative probabilities of COVID-19 outbreak in the nursing homes and those of hospitalization for the disease in the population were modelled in each of the twelve *Départements* of the Region over period March-July 2020. This allowed estimating the duration of the active outbreak period, the dates and heights of the peaks of outbreak probabilities in nursing homes, and the dates and heights of the peaks of hospitalization probabilities in the population. Spearman coefficient estimated the correlation between the two peak series.

### Results

The cumulative proportion of nursing homes with COVID-19 outbreaks was 52% (490/943; range: 22–70% acc. Département). The active outbreak period in the nursing homes lasted 11 to 21 days (acc. Département) and ended before lockdown end. Spearman correlation

**Funding:** The study was funded by the Agence Nationale de la Recherche (Flash COVID-19 program, ANR-20-COVI-000) and the Fondation de France (Engagement 105969). The funders had no role in the study design; data collection and analysis; the preparation of the manuscript; or the decision to publish it.

**Competing interests:** The authors have declared that no competing interests exist.

between outbreak probability peaks in nursing homes and hospitalization probability peaks in the population (surrogate of the incidence peaks) was estimated at 0.71 (95% CI: [0.66; 0.78]).

## Conclusion

The modelling highlighted a strong correlation between the outbreak in nursing homes and the external pressure of the disease. It indicated that avoiding disease outbreaks in nursing homes requires a tight control of virus spread in the surrounding populations.

## Introduction

In December 2019, a severe respiratory syndrome due to a novel coronavirus named subsequently Severe Acute Respiratory Syndrome CoronaVirus 2 (SARS-CoV-2) was identified in Wuhan, China [1, 2]. The disease named COronaVIrus Disease 2019 (COVID-19) has then spread worldwide and, on 11 March 2020, the World Health Organization declared the SARS-Cov-2 outbreak a pandemic. The main mode of transmission of the virus is via the droplets expelled during face-to-face exposure [3]. Prolonged unprotected exposure to symptomatic patients presents the highest risk of transmission. However, pre-symptomatic and asymptomatic individuals can also transmit the virus and are major contributors to the spread of the disease. The most common COVID-19 symptoms are fever, dry cough, shortness of breath, fatigue, digestive symptoms, and myalgia. Anosmia or ageusia are also frequent and can be the sole symptoms. Among the factors associated with the disease severity such as diabetes, hypertension, or obesity, advanced age seems to be a major prognostic factor.

In 2020, elderly people were particularly affected by COVID-19. Worldwide, during the first wave of the pandemic, COVID-19 outbreak levels were notably high in nursing homes. This was reported by a number of large studies. In a large-scale survey carried out in England between May 26 and June 20, 2020, nearly half of nursing homes reported at least one confirmed case [4]. By June 30, 2020, among 13,709 nursing homes of the USA, 39% reported at least one case [5]; this proportion reached even 60% in 1,146 nursing homes of Massachusetts, Georgia, and New Jersey [6]. By May 2020, across Europe, long-term care facilities (that include nursing homes) registered 26 to 66% of all COVID-19 deaths [7]. By November 2020, in France, 44% of COVID-19 deaths occurred in nursing homes and, in these homes, the mean COVID-19 case fatality rate was 20% [8]. Similar high fatality rates were observed in nursing homes of other European countries [9, 10].

During the first epidemic wave, the surge and spread of the disease in nursing homes were favoured by the presence of asymptomatic and pre-symptomatic residents or persons from surrounding communities (staff, visitors, etc.) [10–14] and further aggravated by shortage of protective equipment, tests, and staff [15–17]. This led to recommend testing firstly residents and staff with suggestive symptoms and then contact persons rather than carrying out systematic screening [18]. Thus, the outbreak of COVID-19 in nursing homes might be strongly linked with the spread of the coronavirus in the surrounding population.

To check this hypothesis, we modelled during the first epidemic wave the extent and the dynamics of the disease in the nursing homes of all *Départements* of Auvergne-Rhône-Alpes (ARA) Region. The change in the hospitalization probability in same *Départements'* populations was also modelled over the same period to assess the link between COVID-19 outbreak in the nursing homes and the external pressure of the disease.

**Table 1. COVID-19 cumulative outbreaks in nursing homes and cumulative incidence of inhabitant hospitalizations for COVID-19 in the *Départements* of Auvergne-Rhône-Alpes Region, France, March 1–July 31, 2020.**

| *Département* | Nursing homes | | | Population | | |
| --- | --- | --- | --- | --- | --- | --- |
| | Total number | Nursing homes with outbreak | Cumulative percentage % | Inhabitants[a] | Cumulative incidence of hospitalizations[b] | Cumulative incidence of hospitalizations per 100,000 |
| Cantal | 40 | 9 | 22 | 142,811 | 73 | 51 |
| Allier | 48 | 17 | 35 | 331,315 | 265 | 80 |
| Puy-de-Dôme | 99 | 38 | 38 | 660,240 | 264 | 40 |
| Savoie | 57 | 26 | 46 | 432,548 | 515 | 119 |
| Drôme | 71 | 35 | 49 | 520,560 | 730 | 140 |
| Loire | 112 | 57 | 51 | 764,737 | 1,391 | 182 |
| Haute-Loire | 49 | 26 | 53 | 226,901 | 158 | 70 |
| Isère | 108 | 57 | 53 | 1,264,979 | 954 | 75 |
| Ardèche | 65 | 38 | 58 | 326,875 | 674 | 206 |
| Ain | 67 | 41 | 62 | 656,955 | 640 | 97 |
| Rhône | 161 | 100 | 62 | 1,876,051 | 4,488 | 239 |
| Haute-Savoie | 66 | 46 | 70 | 828,405 | 1,084 | 131 |
| All Dept. | 943 | 490 | 52 | 8,032,377 | 11,236 | 140 |

[a] Source INSEE projections for 2020.

[b] Number of inhabitants hospitalized for COVID-19 during the study period.

## Methods

### Characteristics of ARA Region

ARA Region is the second most important of the 13 Regions of Metropolitan France in terms of population (8,032,377 inhabitants in 2020, 12% of the French population)). It includes twelve Départements whose populations range between 142,811 (Cantal) and 1,876,051 people (Rhône, Département where Lyon metropole is home to nearly 1,400,000 people) (Table 1, Fig 1).

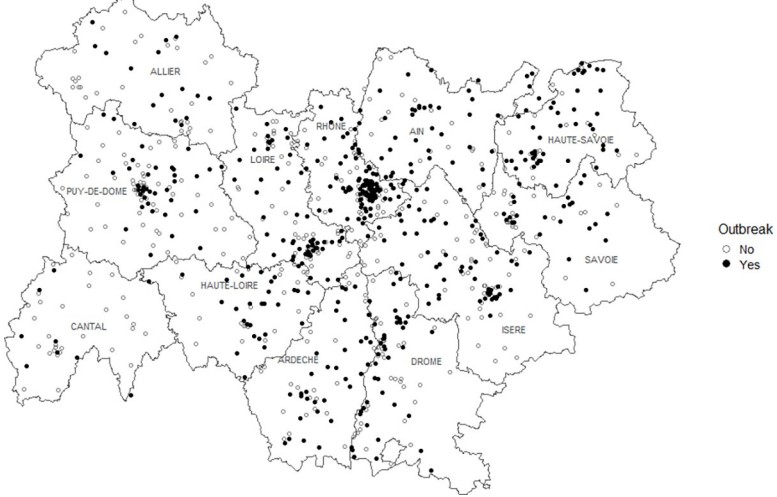

**Fig 1. Locations of the nursing homes of Auvergne-Rhône-Alpes Region, France, March 1–July 31, 2020.** Black and white circles indicate, respectively, the nursing homes with and without COVID-19 outbreaks.

### Nursing home characteristics and collected data

The residents of the nursing homes under study were people aged $\geq$60 years with low to high degree of dependency. In those homes, care is provided by nurses and care assistants under the supervision of a coordinating physician.

The study period was between March 1 and July 31, 2020; this corresponded roughly to the duration of the first epidemic wave.

The data on the epidemic outbreak were extracted from a database designed by Santé Publique France to monitor the surge and spread of COVID-19 in all medico-social facilities, including nursing homes (Cf. https://www.gouvernement.fr/info-coronavirus/carte-et-donnees). This nationwide surveillance system of COVID-19 cases required the declaration of a suspected or confirmed case and then the provision of daily data on the number of cases, deaths, etc.

An outbreak of the disease in a nursing home was declared as soon as the first suspected or confirmed case in a resident or staff member was reported to the surveillance system [19, 20]. The date of this outbreak was that of symptom onset in that first case. A suspected case was defined by the presence of fever, respiratory symptoms, or other clinical symptoms considered by a physician as compatible with COVID-19. A confirmed case was defined as one with a positive SARS-CoV-2 RT-PCR test result.

### Imputation of missing dates of symptom onsets in first cases

The date of symptom onset in the first case was missing in 44% of the 490 nursing homes that reported COVID-19 cases during the study period. For each nursing home with missing date, a nearest neighbour imputation algorithm was developed to identify the ten nearest homes in terms of date of first report, number of reported cases, and number of deaths. The delay between symptom onset in the first case and the date of first report was sampled from the ten nearest neighbours and used to impute the outbreak date. This sampling was repeated ten times to build ten datasets for analysis.

### Hospitalization data

The daily number of hospitalized persons in each *Département* of ARA Region was extracted from the national database on COVID-19-related hospitalizations (https://www.gouvernement.fr/info-coronavirus/carte-et-donnees). The population of each *Département* was obtained from projections for 2020 of national census data (https://www.insee.fr/fr/statistiques/2859843).

### Characteristics of the outbreaks in the nursing homes and the population

The outbreaks were characterized by the duration of the active outbreak period, the date and height of the peak of outbreak probability in nursing homes, and the date and height of the peak of hospitalization probability in the population. Estimating these characteristics required modelling the observed data.

### Modelling

The cumulative proportions of COVID-19 outbreaks in the nursing homes over the study period were modelled in each *Département* using the five-parameter non-linear model of Brain and Cousens (an extension of the four-parameter logistic model) [21]. The first and second derivatives of each predicted cumulative probability curve allowed extracting: i) T1: the delay to acceleration of the probability increase; ii) T2: the delay to the probability peak; and, iii) T3: the delay to deceleration of the probability decrease (Fig 2).

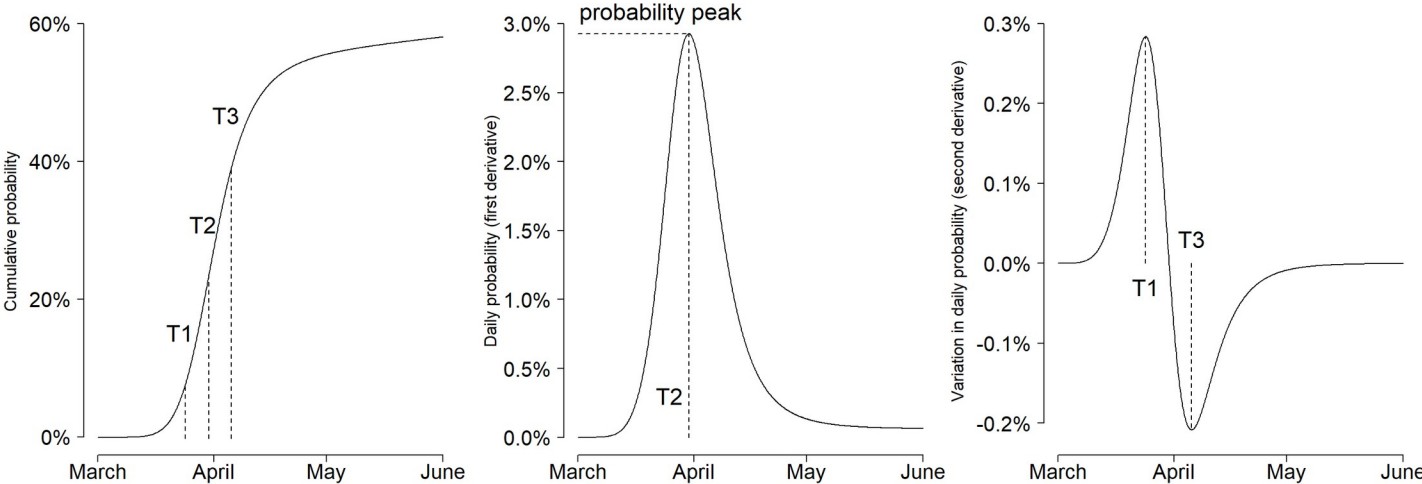

**Fig 2. Curve of cumulative probabilities of COVID-19 outbreak in the nursing homes of *Département* Ain, Auvergne-Rhône-Alpes Region, France, March 1–June 31, 2020.** T1: delay to acceleration of the probability increase. T2: delay to the peak of outbreak probability. T3: delay to deceleration of the probability decrease, expressed in days since March 1, 2020.

T1, T2, and T3 were expressed in number of days elapsed since March 1, 2020. The duration of the active outbreak period was calculated as the lag time (in days) between T1 and T3. The mean cumulative curves were built using the mean of parameter estimates over the ten datasets (S1A Fig). A sampling from the distribution of each parameter estimates was used to determine the 95% confidence interval of each indicator. The same analysis strategy was used to model the cumulative hospitalization probabilities (S1B Fig).

The correlation between the ranks of the peaks of outbreak probability in nursing homes and the ranks of the peaks of hospitalization probability was estimated using Spearman correlation coefficient.

The analysis was carried out with R statistical software, version 3.6.1. [22].

## Ethics

The study was conducted in agreement with the European General Data Protection Regulation and approved by the institutional research ethics committee (Comité d'Éthique du Centre Hospitalier Universitaire de Lyon, N° 20–81).

The use of anonymized and aggregated data obviated the need for participants' consents.

## Results

The cumulative proportion of nursing homes of ARA Region that reported COVID-19 outbreaks was 52% (490 / 943; range: 22–70% according to the *Département*) (Table 1, Fig 3). The cumulative proportion of hospitalizations for COVID-19 in the Region was 140 per 100,000 inhabitants (range: 40–239 according to the *Département*).

The active outbreak period in the nursing homes (i.e., T1 to T3) lasted 11 to 21 days, according to the *Département*, and ended before the end of the lockdown on May 11 (Table 2, Fig 3, and S1C Fig).

The peak of the outbreak probability in the nursing homes occurred 7 to 17 days after the beginning of the lockdown (i.e., March 15) and before the peak of hospitalization probability in all *Départements* but one (with two to nine days delay between the two peaks) (Table 2, Fig 3). The geographic distribution of the outbreaks in the nursing homes was heterogeneous,

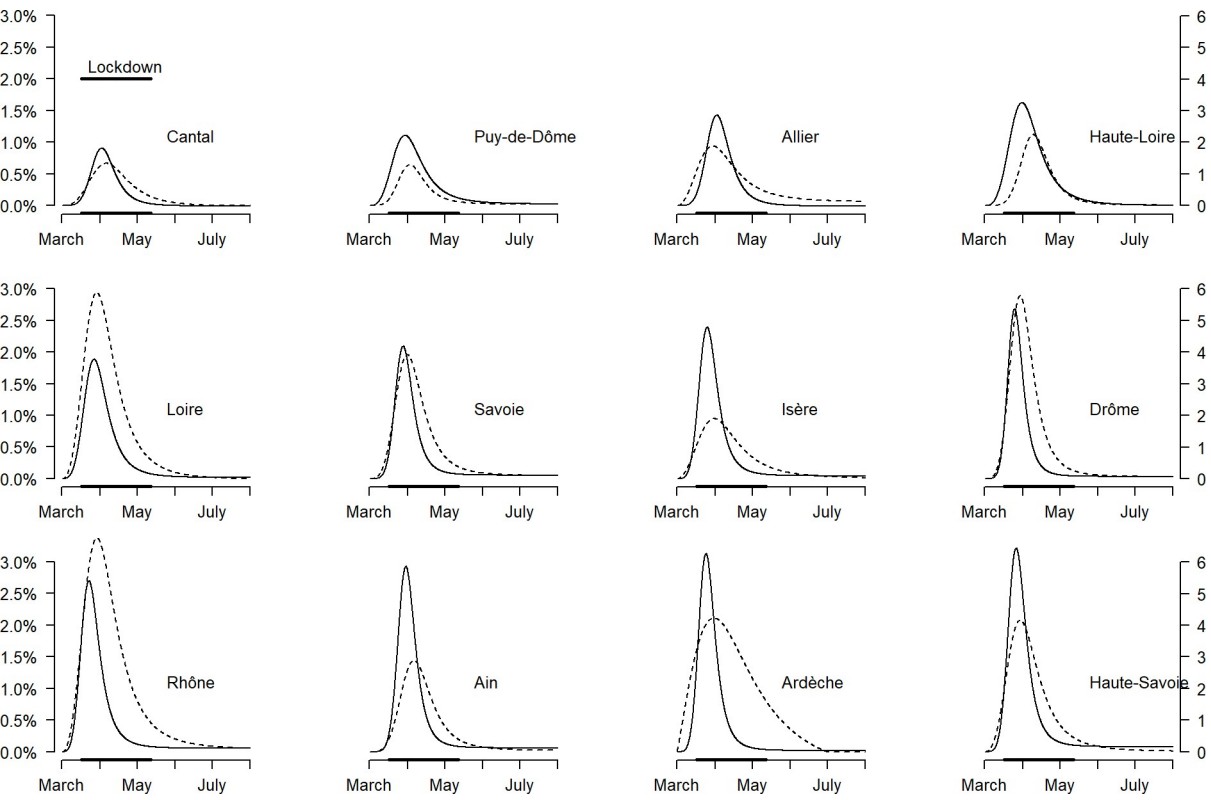

**Fig 3. Modelled curves of daily outbreak probabilities of COVID-19 in the nursing homes of each *Département* (solid line) and daily hospitalization probabilities in each *Département* population (dotted line) according to the time elapsed since March 1, 2020, France.** The curves are ranked in an increasing order of height of the peaks of outbreak probabilities in nursing homes. The left-hand y-axis corresponds to the daily probability of outbreak. The right-hand y-axis corresponds to the daily hospitalization probability per 100,000 inhabitants.

especially at the beginning of the first epidemic wave (S1C Fig). The peak of hospitalization probabilities occurred between March 28 and 31 in eight *Départements* out of twelve. In the four remaining *Départements* with hospitalization probability peaks occurring after March 31, the heights of the peaks were lower (Table 2).

The correlation between the ranks of the *Départements* according to the outbreak probability peak in the nursing homes and their ranks according to the hospitalization probability peak was estimated at 0.71 (95% CI: [0.66; 0.78]). This indicated a strong association between COVID-19 outbreak in the nursing homes and the spread of the disease in the surrounding population (Fig 4).

The four *Départements* with the lowest outbreak probability peaks in nursing homes were also the *Départements* with the lowest hospitalization probability peaks; three of which had the lowest population densities. However, *Département* Rhône (that includes Lyon metropole) that had the highest hospitalization probability peak and population density was not the one that had the highest outbreak probability peak in nursing homes. In *Département* Ardèche (that has a low population density), the outbreak and hospitalization probability peaks were high. In *Départements* Isère and Ain, the outbreak probability peaks were high, whereas the hospitalization probability peaks were low (Fig 3). The duration of the active outbreak period was shorter in *Départements* with outbreak probability peaks > 2% than in other *Départements* (Table 2).

**Table 2. Estimated durations of the active periods of COVID-19 outbreaks in nursing homes, estimated delays and values of the outbreak probability peaks in the nursing homes and the hospitalization probability peaks for COVID-19 in the populations of Auvergne-Rhône-Alpes Region *Départements*, France, March 1–July 31, 2020.**

| *Département* | Outbreaks in nursing homes | | | Hospitalizations in *Département* populations | | |
|---|---|---|---|---|---|---|
| | Active period duration[a] (T1-T3) | Delay to the peak since March 1, 2020 (T2) | Value of the peak % | Delay to the peak since March 1, 2020 (T2) | Value of the peak, per 100,000 inhabitants | Population density[b] |
| Cantal | 17 [15;21] | 32 [30;34] | 0.91 [0.76;1.01] | 36 [35;37] | 1.34 [1.29;1.39] | 24.9 |
| Puy-de-Dôme | 21 [19;24] | 29 [27;30] | 1.11 [1.02;1.19] | 33 [32;34] | 1.29 [1.23;1.36] | 82.8 |
| Allier | 17 [14;20] | 32 [29;34] | 1.43 [1.23;1.62] | 29 [27;31] | 1.88 [1.80;1.97] | 45.1 |
| Haute Loire | 21 [18;25] | 30 [28;32] | 1.62 [1.46;1.83] | 39 [39;40] | 2.24 [2.16;2.33] | 51.7 |
| Loire | 17 [16;18] | 26 [25;27] | 1.89 [1.77;1.98] | 28 [28;29] | 5.88 [5.80;5.96] | 160.0 |
| Savoie | 13 [11;14] | 27 [26;29] | 2.09 [1.90;2.43] | 31 [30;32] | 3.92 [3.81;4.02] | 71.8 |
| Isère | 13 [11;15] | 24 [23;25] | 2.39 [2.17;2.71] | 30 [29;31] | 1.90 [1.87;1.92] | 170.2 |
| Drôme | 11 [9;13] | 24 [22;26] | 2.68 [2.31;3.08] | 29 [28;29] | 5.78 [5.69;5.87] | 79.7 |
| Rhône | 14 [12;21] | 22 [21;23] | 2.71 [2.56;2.94] | 28 [28;29] | 6.76 [6.68;6.85] | 577.4 |
| Ain | 12 [10;14] | 30 [29;31] | 2.93 [2.53;3.63] | 36 [36;37] | 2.89 [2.83;2.95] | 114.0 |
| Ardèche | 12 [10;14] | 24 [23;24] | 3.13 [2.72;3.78] | 30 [22;34] | 4.23 [3.93;4.56] | 59.1 |
| Haute Savoie | 13 [11;14] | 26 [25;27] | 3.22 [2.90;3.53] | 29 [28;29] | 4.16 [4.10;4.21] | 188.8 |

Whenever applicable, values are followed by their [95% confidence intervals].

The durations and the delays are expressed in number of days.

[a] Delay between the date of acceleration of outbreak probability increase (T1) and the date of deceleration of outbreak probability decrease (T3).

[b] Number of inhabitants per km$^2$ according to the French INSEE projections for 2020.

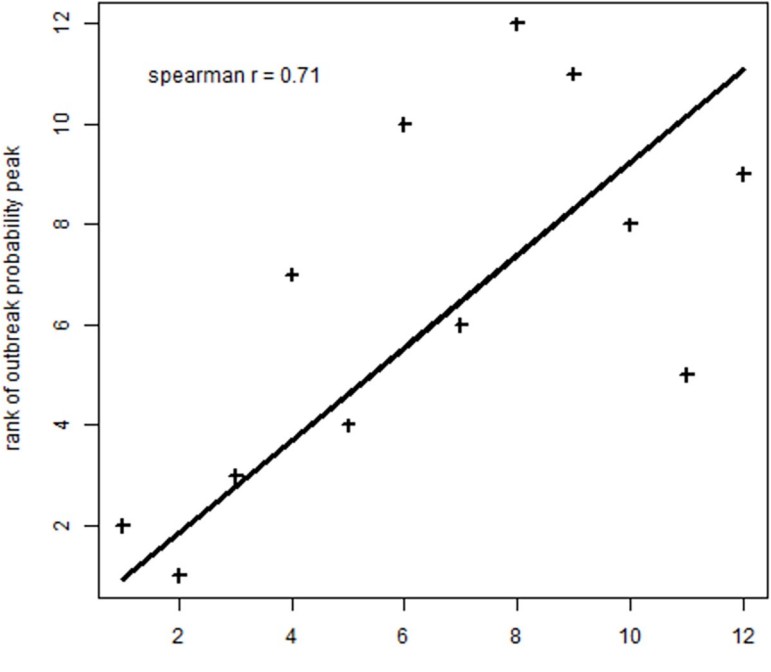

**Fig 4. Correlation between *Département* ranks according to the outbreak probability peak in nursing homes and *Département* ranks according to the hospitalization probability peak in the population in the Auvergne-Rhône-Alpes Region, France, March 1–July 31, 2020.**

## Discussion

In the French ARA Region, as in other settings, the cumulative proportion of nursing homes that reported COVID-19 outbreaks during the first epidemic wave was high [4–6, 23]. Fifty-two percent of nursing homes declared at least one suspected or confirmed case in a resident or staff member. However, there was an important heterogeneity between *Départements* that may be explained by the heterogeneity of the disease incidence in the *Départements*' populations. Indeed, the analyses showed a strong positive correlation between the ranks of the outbreak probability peaks in the nursing homes of the *Départements* and the ranks of the hospitalization probability peaks in the same *Départements* (the latter being indirect measurements of incidence peaks). This correlation was already mentioned in other studies [5, 6, 24–26]. In a study by Sun et al. [6], in 1,146 nursing homes of Massachusetts, Georgia, and New Jersey, the strongest predictors of the probability of presence of at least one COVID-19 case were the infection rate in the county and the number of care units in the nursing homes. A third predictor was the population density, which was positively linked with the outbreak probability in the nursing homes. According to Sugg et al. [5], COVID-19 transmission in 13,709 nursing homes in the USA was positively associated with the corresponding county infection rate and the population density. Similarly, the present study showed that the *Départements* with the lowest peaks of hospitalization probability were also the *Départements* with the lowest population densities. Another indicator of the strong link between coronavirus spread in the general population and the risk of disease outbreak in nursing homes was the concurrent strong negative impacts of the lockdown on the hospitalization probability in the population and on the outbreak probability in the nursing homes.

The study results support the following recommendations for protecting the vulnerable residents in nursing homes: i) reduce the virus spread in the general population; ii) test regularly all staff members of nursing homes located in areas with high levels of virus spread in the population (Cf. the ECDC guide for surveillance and control in long-term care facilities) [7]; iii) acquire and use personal protective equipment; and, iv) comply with the universal barrier measures soon after an outbreak alert in the general population.

In addition, the study found that, in most *Départements*, the peak of outbreak probability in nursing homes preceded the peak of hospitalization probability. This suggests that a decreasing risk of outbreak in nursing homes might indicate a shift toward a decrease in hospitalization rates. This shift may be used to guide hospital bed management.

### Limitations of the study

First, the incidence of hospitalization allowed an indirect quantification of the disease incidence in the general population (obtained with a 13.5 day delay, on average) [27]. In each *Département*, as the delay between the peak of outbreak probability in the nursing homes and the peak of hospitalization probability in the surrounding population was estimated at two to nine days, the peak in the nursing homes should closely follow the peak of disease incidence in the population. Second, the incidence of hospitalization at a *Département* level was an imperfect measure of the extent of the virus spread in the surrounding populations. This may explain the discrepancy between the heights of the outbreak probability peak in the nursing homes and the hospitalization probability peak in some *Départements* (Isère or Ain). Third, the study did not explore other characteristics of the nursing homes that could be linked with the outbreak probability. This will be the object of a future analysis.

In conclusion, this study highlighted that the outbreak of COVID-19 in nursing homes occurred soon after its outbreak in the general population and was highly correlated with the incidence of the disease in the surrounding populations. Consequently, avoiding or limiting

outbreaks of the disease in nursing homes requires a tight control of virus spread in the surrounding populations and a quick implementation of virus screening and barrier measures in those homes soon after outbreak alerts in the nearby populations.

## Supporting information

**S1 Fig. Observed and modelled cumulative curves of COVID-19 outbreaks in the nursing homes of each *Département* of Auvergne-Rhône-Alpes Region over the study period (March 1 –July 31, 2020).**
(PDF)

**S2 Fig. Observed and modelled cumulative curves of hospitalization for COVID-19 in the *Départements* of Auvergne-Rhône-Alpes Region over the study period (March 1 –July 31, 2020).**
(PDF)

**S3 Fig. Weekly dynamics of COVID-19 outbreaks in the nursing homes of Auvergne-Rhône-Alpes Region over the study period (943 nursing homes; March 1 –July 31, 2020).**
(PDF)

**S1 Dataset. This file in csv format contains the data allowing to model the outbreak in the nursing homes of each *Département*.** Description of the dataset: 943 lines (one line per nursing home) and 3 columns (Number, *Département*, DateOutbreak). Number: number of the nursing home. *Département*: number of the *Département*. DateOutbreak: date of the beginning of the outbreak.
(CSV)

**S2 Dataset. This file in csv format contains the data allowing to model the hospitalization incidence in each *Département*.** Description of the dataset: 135 lines per *Département* and four columns (*Département*, DateHospitalization, NumberHosp, NumberPopDpt). *Département*: number of the *Département*. DateHospitalization: Date of hospitalization. NumberHosp: number of inhabitants hospitalized the given day. NumberPopDpt: population size of the *Département*.
(CSV)

## Acknowledgments

The authors would like to thank the healthcare professionals of the nursing homes in Auvergne-Rhône-Alpes Region who helped carrying out the survey and the agents of Agence Régionale de Santé Auvergne-Rhône-Alpes who were involved in the implementation of the control measures.

The authors also thank the following epidemiologists involved in the surveillance system: Delphine Casamatta, Philippe Pépin, Christine Saura, Garance Terpant, Mélanie Yvroud (Santé Publique France Auvergne-Rhône-Alpes), Kostas Danis, Scarlett Georges, Côme Daniau, Laure Fonteneau (Département des maladies infectieuses, Santé Publique France).

## Author Contributions

**Conceptualization:** Muriel Rabilloud, Jean François Etard, Nicolas Voirin, René Ecochard, Philippe Vanhems.

**Formal analysis:** Muriel Rabilloud, Benjamin Riche, Mad-Hélénie Elsensohn.

**Funding acquisition:** Muriel Rabilloud, Jean François Etard, Nicolas Voirin, René Ecochard, Philippe Vanhems.

**Investigation:** Thomas Bénet.

**Methodology:** Muriel Rabilloud, Benjamin Riche, Mad-Hélénie Elsensohn, René Ecochard.

**Supervision:** Muriel Rabilloud, Philippe Vanhems.

**Writing – original draft:** Muriel Rabilloud, Benjamin Riche, Jean Iwaz, René Ecochard.

**Writing – review & editing:** Muriel Rabilloud, Benjamin Riche, Jean François Etard, Mad-Hélénie Elsensohn, Nicolas Voirin, Thomas Bénet, Jean Iwaz, René Ecochard, Philippe Vanhems.

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
