## [Decision Letter · Decision Letter 0]

4 Aug 2021

PONE-D-21-20188

COVID-19 outbreaks in nursing homes: a strong link with the coronavirus spread in the surrounding population, France, March to July 2020

PLOS ONE

Dear Dr. Rabilloud,

Thank you for submitting your manuscript to PLOS ONE. After careful consideration, we feel that it has merit but does not fully meet PLOS ONE’s publication criteria as it currently stands. Therefore, we invite you to submit a revised version of the manuscript that addresses the points raised during the review process.

We look forward to receiving your revised manuscript.

Kind regards,

Giordano Madeddu

Academic Editor

PLOS ONE

Journal Requirements:

“PV, MR, RE, JFE, NV

ANR-20-COVI-000

Agence Nationale de la Recherche, programme Flash COVID-19

Engagement 105969

Fondation de France”

4. We note that Figure 1 and S3 Figure in your submission contain map images which may be copyrighted. All PLOS content is published under the Creative Commons Attribution License (CC BY 4.0), which means that the manuscript, images, and Supporting Information files will be freely available online, and any third party is permitted to access, download, copy, distribute, and use these materials in any way, even commercially, with proper attribution. For these reasons, we cannot publish previously copyrighted maps or satellite images created using proprietary data, such as Google software (Google Maps, Street View, and Earth). For more information, see our copyright guidelines: http://journals.plos.org/plosone/s/licenses-and-copyright.

a. You may seek permission from the original copyright holder of Figure 1 and S3 to publish the content specifically under the CC BY 4.0 license. 

Reviewers' comments:

Reviewer's Responses to Questions

**Comments to the Author**

1. Is the manuscript technically sound, and do the data support the conclusions?

Reviewer #1: Yes

Reviewer #2: Yes

2. Has the statistical analysis been performed appropriately and rigorously? 

Reviewer #1: Yes

Reviewer #2: Yes

3. Have the authors made all data underlying the findings in their manuscript fully available?

Reviewer #1: No

Reviewer #2: Yes

4. Is the manuscript presented in an intelligible fashion and written in standard English?

Reviewer #1: Yes

Reviewer #2: Yes

5. Review Comments to the Author

Reviewer #1: Rabilloud et al aimed to evaluate the link between COVID-19 in nursing homes and SARS.CoV.2 spread on surrounding population.

There is a rising interest on nursing homes, giving their particular settings and their relationship with SARS-CoV-2 spread. It follows the paper focus is interesting and suitable to be shared with the scientific community.

However, there are some points to address before the paper would be ready for publication.

Introduction

Introduction is quite short and would benefit from generalities adding before introducing nursing homes importance. Please, to make it more complete, add some more data on virus importance and most frequent clinical features. As example:

- Generalities: In December 2019, a new severe respiratory syndrome was identified in Wuhan, China. On January 2020, a new Coronavirus was detected and called SARS-CoV-2. On March 2021, the WHO declared COVID-19 as a public health emergency. (Commission WMH. Available: http://wjw.wuhan.gov.cn/front/web/showDetail/2019123108989; New-type coronavirus causes pneumonia in Wuhan: expert - Xinhua | English.news.cn. Available: http://www.xinhuanet.com/english/2020-01/09/c_138690570.htm)

- Pathophysiology and transmission: e.g. https://doi.org/10.1186/s40779-020-00240-0; https://doi.org/10.1001/jama.2020.12839;
https://doi.org/10.26355/eurrev_202101_24424

- Most common clinical features: e.g. https://doi.org/10.26355/eurrev_202007_22291;
https://doi.org/10.1002/hed.26269;
https://doi.org/10.1016/S1473-3099(20)30402-3.

Then, introduce nursing homes importance for SARS-CoV-2 spread.

Methods

- May you move data on ARA Region from Methods to Introduction? It is just descriptive and is not exactly a ‘Method’. Furthermore, with a slightly better description and references the table could be avoided.

- Study population. This should be slightly better defined. It is not clear the level of medical/nursing assistance needed in the setting (low, medium, high level of patient’s dependency). Are they sheltered care, residential care home residents, or nursing home residents?. It would be useful to better understand the settings and, as consequence, the level of contacts with visitors and healthcare providers.

Discussion

Discussion seems to be well constructed. However, there are at least other two recent manuscripts regarding nursing home in the same Journal. Please, check and comment: https://doi.org/10.1371/journal.pone.0255141;
https://doi.org/10.1371/journal.pone.0248009

Limitations

Please, provide limitations in a paragraph apart with ‘Limitations of the study’ heading.

Language

Language should be slightly revised, in order to make the paper more clear (e.g. some parts may be rephrased to make the sentence even shorter).

In conclusion, with some modifications, the article will be an added value for the actual knowledge on SARS-CoV-2.

Reviewer #2: Title: SARS due to COVID-19: predictors of death and profile of patients in the state of Rio de Janeiro, 2020 Manuscript number PONE-D-21-19718

Review by Mastewal Arefaynie /Assistant professor in public health)

Wollo University

Dessie, Ethiopia

General comment

There are several topological and grammar usage errors that need extensive proof reading for revisions.

Specific comments

Abstract

1. In the introduction part you state simply the objective of the study. But it needs the justification of the research (the identified gap).

2. Material and method part it is enough to say method. So remove material. Try to include the software you used for analysis and the type logistic regression you were used.

3. Result: are you using all SARS case or COVID-19 patients only? Try to focus only on the latter case.

4. Line “32” comorbidity was risk factor for death. But you state comorbidity like kidney disease in line “32-34”. But preferred to use each comorbidity factor by removing their comorbidity.

5. Immunodepression change to immunosuppression

6. The conclusion part Line “36” by using odds ratio, you try to conclude to factors. But it is advisable to include more predictors with direction.

Introduction

Generally it is good. But need some justification.

7. The justification to do the research is not well described for scholars.

Methods

8. Change “Materials and Methods” to “methods”

9. “Line 82”, immediate notification of SARS cases was. Change cases to case or was to were.

10. “Line 86” during your description of the outcome variable, you say none death for hospitalization, and hospitalization in an intensive care unit patients. Why not declare their outcome? Unless miss-classification may be there. Your dependent variable should be cure and death.

11. You are using general linear logistic regression. But your outcome variable is dichotomous so binary logistic regression is appropriate. Also your result is expressed in OR.

Result

Unless you were doing comparative study among COVID-19 SARS and SARS, write the result only for you interest. Or compare them.

Discussion

Needs justification for factors

Wish you the luckiest!

Mastewal Arefaynie.

6. PLOS authors have the option to publish the peer review history of their article (what does this mean?). If published, this will include your full peer review and any attached files.

Reviewer #1: No

Reviewer #2: No

---

## [Author Response · Author response to Decision Letter 0]

1 Oct 2021

Dear Sir or Madam,

Please find below the answers to the points raised by the Academic Editor and Reviewer 1 regarding our manuscript untitled “COVID-19 outbreaks in nursing homes: a strong link with the coronavirus spread in the surrounding population, France, March to July 2020”.

In an e-mail sent on 23 August, we have informed the editorial office of the journal that the comments of Reviewer 2 do concern another manuscript written by another team (precisely, manuscript PONE-D-21-19718 “SARS due to COVID-19: predictors of death and profile of patients in the state of Rio de Janeiro, 2020). After your reply on 24 August, we received no other review and considered there were no additional comments to deal with.

We hope the following answers will prove satisfactory and that the revised manuscript will be soon accepted.

Best regards,

Muriel Rabilloud

Answers to the Academic Editor comments

We have made these checks.

“PV, MR, RE, JFE, NV

ANR-20-COVI-000

Agence Nationale de la Recherche, programme Flash COVID-19

Engagement 105969

Fondation de France”

The statement was checked and the lack of roles of the funders added to the main text (See section ‘Funding’): “The funders had no role in the study design; data collection and analysis; the preparation of the manuscript; or the decision to publish it.”

We have prepared twol files in csv format for the data that allow modeling respectively the outbreaks in nursing homes and incidence of hospitalization for COVID-19 in each of the twelve Départements of Auvergne-Rhône-Alpes Region. The first file consists of 942 lines (one per nursing home) and three columns (number of nursing homes, Département number, and date of outbreak). The second file consists of 135 lines per Département and four columns (Département number, date, number of hospitalisations, population size). The files will be submitted with the revised manuscript as Supporting Information files.

4. We note that Figure 1 and S3 Figure in your submission contain map images which may be copyrighted. All PLOS content is published under the Creative Commons Attribution License (CC BY 4.0), which means that the manuscript, images, and Supporting Information files will be freely available online, and any third party is permitted to access, download, copy, distribute, and use these materials in any way, even commercially, with proper attribution. For these reasons, we cannot publish previously copyrighted maps or satellite images created using proprietary data, such as Google software (Google Maps, Street View, and Earth). For more information, see our copyright guidelines: http://journals.plos.org/plosone/s/licenses-and-copyright.

Figures 1 and S3 are not copyrighted. We built the maps of Auvergne-Rhône-Alpes Region using R software and the GPS coordinates of the nursing homes. 

We have reviewed the reference list and added five references in response to the reviewer’s queries (see below). 

Answers to Reviewer 1 comments

We thank Reviewer 1 for his interest in our work and hope the following answers will prove satisfactory. 

Introduction

Introduction is quite short and would benefit from generalities adding before introducing nursing homes importance. Please, to make it more complete, add some more data on virus importance and most frequent clinical features.

As required by the reviewer, we have added generalities on the pandemic due to SARS-Cov-2 infection (+ three references) before introducing the importance of the disease in the nursing homes (lines 27 to 39). 

Methods

May you move data on ARA Region from Methods to Introduction? It is just descriptive and is not exactly a ‘Method’. Furthermore, with a slightly better description and references the table could be avoided.

The authors have examined the point and deemed Table 1 can hardly be avoided because it presents the observed data used to model the outbreaks in the nursing homes and hospitalization incidence. Thus, they believe these data may remain in section ‘Methods’ because they show the material and the setting of the study.

- Study population. This should be slightly better defined. It is not clear the level of medical/nursing assistance needed in the setting (low, medium, high level of patient’s dependency). Are they sheltered care, residential care home residents, or nursing home residents? It would be useful to better understand the settings and, as consequence, the level of contacts with visitors and healthcare providers.

We have added information on the characteristics of the residents in the nursing homes and on the type of care provided in these facilities (lines 96 to 99).

Discussion

Discussion seems to be well constructed. However, there are at least other two recent manuscripts regarding nursing home in the same Journal. Please, check and comment: https://doi.org/10.1371/journal.pone.0255141;
https://doi.org/10.1371/journal.pone.0248009

The above-cited manuscripts are now mentioned in the Introduction as supplementary references highlighting the severity of the COVID-19 for the residents of nursing homes (references 9 and 10, lines 51-52).

The aim of the paper of De Vito et al. was to assess the spread of the infection in nursing homes where at least one infected person was present, and identify the predictors of developing symptoms and die. The aim of the paper of Meis-Pinheiro et al. was to describe the clinical characteristics and the prognosis of the disease in a cohort of residents of long-term nursing homes who were infected during the first wave of the pandemic. One aim of this study was also to assess the link between the organization of the nursing homes and the incidence of the disease. However, these two studies did not explore the link between the outbreak of COVID-19 in nursing homes and the spread of the coronavirus in the surrounding population. Thus, the two papers present interesting results but we did not feel we have to include them in our Discussion. 

Limitations

Please, provide limitations in a paragraph apart with ‘Limitations of the study’ heading.

We mention now the limitations of the study in a separate paragraph. 

Language

Language should be slightly revised in order to make the paper more clear (e.g., some parts may be rephrased to make the sentence even shorter).

The whole text has been revised to clarify or simplify the expression of long sentences.

---

## [Decision Letter · Decision Letter 1]

10 Dec 2021

COVID-19 outbreaks in nursing homes: a strong link with the coronavirus spread in the surrounding population, France, March to July 2020

PONE-D-21-20188R1

Dear Dr. Rabilloud,

We’re pleased to inform you that your manuscript has been judged scientifically suitable for publication and will be formally accepted for publication once it meets all outstanding technical requirements.

Kind regards,

Giordano Madeddu

Academic Editor

PLOS ONE

Additional Editor Comments (optional):

Reviewers' comments:

Reviewer's Responses to Questions

**Comments to the Author**

1. If the authors have adequately addressed your comments raised in a previous round of review and you feel that this manuscript is now acceptable for publication, you may indicate that here to bypass the “Comments to the Author” section, enter your conflict of interest statement in the “Confidential to Editor” section, and submit your "Accept" recommendation.

Reviewer #1: All comments have been addressed

2. Is the manuscript technically sound, and do the data support the conclusions?

Reviewer #1: Yes

3. Has the statistical analysis been performed appropriately and rigorously? 

Reviewer #1: Yes

4. Have the authors made all data underlying the findings in their manuscript fully available?

Reviewer #1: Yes

5. Is the manuscript presented in an intelligible fashion and written in standard English?

Reviewer #1: Yes

6. Review Comments to the Author

Reviewer #1: The authors addressed the most part of my comments and in a satisfying way. I think the paper may actually be published.

7. PLOS authors have the option to publish the peer review history of their article (what does this mean?). If published, this will include your full peer review and any attached files.

Reviewer #1: No

---

## [Editor Report · Acceptance letter]

31 Dec 2021

PONE-D-21-20188R1 

COVID-19 outbreaks in nursing homes: a strong link with the coronavirus spread in the surrounding population, France, March to July 2020 

Dear Dr. Rabilloud:

I'm pleased to inform you that your manuscript has been deemed suitable for publication in PLOS ONE. Congratulations! Your manuscript is now with our production department. 

Kind regards, 

on behalf of

Dr. Giordano Madeddu 

Academic Editor

PLOS ONE